# Genome-Wide Identification and Expression of the *Paulownia fortunei* MADS-Box Gene Family in Response to Phytoplasma Infection

**DOI:** 10.3390/genes14030696

**Published:** 2023-03-11

**Authors:** Minjie Deng, Yang Dong, Saisai Xu, Shunmou Huang, Xiaoqiao Zhai, Guoqiang Fan

**Affiliations:** 1Institute of Paulownia, Henan Agricultural University, Zhengzhou 450002, China; 2College of Forestry, Henan Agricultural University, Zhengzhou 450002, China; 3Henan Academy of Forestry, Zhengzhou 450002, China

**Keywords:** *Paulownia fortunei*, MADS-box gene family, Paulownia witches’ broom, expression profile

## Abstract

Paulownia witches’ broom (PaWB), caused by phytoplasmas, is the most devastating infectious disease of Paulownia. Although a few MADS-box transcription factors have been reported to be involved in the formation of PaWB, there has been little investigation into all of the MADS-box gene family in Paulownia. The objective of this study is to identify the MADS-box gene family in *Paulownia fortunei* on a genome-wide scale and explore their response to PaWB infection. Bioinformatics software were used for identification, characterization, subcellular localization, phylogenetic analysis, the prediction of conserved motifs, gene structures, cis-elements, and protein-protein interaction network construction. The tissue expression profiling of PfMADS-box genes was analyzed by quantitative real-time polymerase chain reaction (qRT-PCR). Transcriptome data and the protein interaction network prediction were combined to screen the genes associated with PaWB formation. We identified 89 MADS-box genes in the *P. fortunei* genome and categorized them into 14 subfamilies. The comprehensive analysis showed that segment duplication events had significant effects on the evolution of the PfMADS-box gene family; the motif distribution of proteins in the same subfamily are similar; development-related, phytohormone-responsive, and stress-related cis-elements were enriched in the promoter regions. The tissue expression pattern of PfMADS-box genes suggested that they underwent subfunctional differentiation. Three genes, *PfMADS3*, *PfMADS57*, and *PfMADS87*, might be related to the occurrence of PaWB. These results will provide a valuable resource to explore the potential functions of PfMADS-box genes and lay a solid foundation for understanding the roles of PfMADS-box genes in paulownia–phytoplasma interactions.

## 1. Introduction

Transcription factors govern transcription initiation by identifying cis-acting regions in the DNA sequence and have been proven to influence plant growth and development, organ morphogenesis, stress response, and hormone signal transduction [1]. The MADS-box gene family is one of the largest transcription factor families in eukaryotes [2]. The MADS-box domain with 56–58 aa in length, found in the N-terminal region of all MADS proteins, recognizes and binds to the CArG motif in the promoters of its target genes [3]. In plants, MADS-box proteins have been divided into two major groups: type I and type II, according to their DNA-binding ability [4]. Type I MADS-box genes consist of four subtypes: Mα, Mβ, Mγ, and Mδ, which generally include 1–2 exons, 0–1 intron, and the encoded proteins contain a conserved SRF-like MADS domain and miss a K domain [3,5]. Type II MADS-box genes typically include seven exons and six introns, with a highly conserved myocyte enhancer factor-2 (MEF2)-like MADS domain and three additional structural domains (a moderately conserved K domain, a conserved I domain, and a variable C terminus) in the encoded proteins [6]. Type Ⅱ MADS-box genes have been further divided into two subtypes, MIKC^C^ and MIKC*, depending on their structure characteristics [6]. MIKC^C^-type genes are the most extensively studied MADS-box genes because of their significant roles in floral organ identification, including the floral homologous genes, and ABCDE model genes [7].

MADS-box genes are notable transcriptional regulators in the ontogeny of higher plants, including Arabidopsis [7], rice [8], wheat [9], tobacco [10], apple [11], and Chinese jujube [12]. MADS-box genes not only regulate floral organ development but also play a vital role in flowering time control, meristem differentiation, embryo development, and formation of roots, seeds, and fruits [12,13,14,15]. *AtFUL* regulates leaf traits by affecting vascular bundle development in Arabidopsis [16]. The cooperation of the MIKC-type gene *AGL21* and several phytohormones could control lateral root development [17]. *NtSOC1* (also known as *NtMADS133*) overexpression promotes early flowering and dwarf symptoms in tobacco [10]. Increased expression of *OsMADS15* causes an early-maturing phenotype characterized by early internode elongation and blossoming, shoot root formation, and decreased plant height [18]. Transgenic wheat plants overexpressing *TaSEP3-1*, a member of the MIKC-type SEP3 subfamily, displayed symptoms of delayed heading stage, reduced plant height, and enhanced tillering [19]. The Paulownia kawakamii plants transformed with *PkMADS1* generated axillary shoots, demonstrating that MADS-box genes play roles in axillary bud establishment [20].

Paulownia is an important fast-growing wood and garden tree. They are widely cultivated in China because of their high adaptability, rapid growth rate, and high-quality wood. They make contributions to food security, environmental improvement, and the enhancement of people’s living standards [21,22]. However, these tree species are susceptible to phytoplasmas that cause Paulownia disease (PaWB) disease, which leads to huge economic losses to the Paulownia industry PaWB results in a series of symptoms [23], such as greening, foliar flowers, axillary bud proliferation, and retarded development. MADS-box genes have recently been shown to involve in the phytoplasma-Paulownia interaction [24,25]. However, to date, the MADS-box gene family has not been determined in Paulownia.

Here, we identified the PfMADS-box gene family in the *Paulownia fortunei (P. fortunei)* genome and analyzed their phylogenetic relationships, gene structure, and conserved protein motifs. Moreover, several MADS-box genes that may be related to PaWB were analyzed using RNA-seq data and protein interaction prediction. These results will lay a foundation for further research on the function of the PfMADS-box gene and provide a clue for an in-depth understanding of the pathogenesis of PaWB.

## 2. Materials and Methods

### 2.1. Plant Materials and Treatments

All the seedlings used in this study were from the Institute of Paulownia, Henan Agricultural University, Zhengzhou, Henan Province, China. Plants were cultivated and treated following the procedures of Fan et al. [26].

In this study, six samples were used to analyze the relationship between the expression of PfMADS-box genes and the formation of PaWB, including healthy *P. fortunei* (PF) seedlings and those infected with phytoplasmas (PFI), diseased seedlings treated with 60 mg·L^−1^ methylmethane sulphonate (MMS) for 10 and 20 days, respectively (PFIM60-10, PFIM60-20), and diseased seedlings treated with 20 mg·L^−1^ MMS for 10 and 30 days, respectively (PFIM20-10, PFIM20-30).

Three tissue samples (flower, stem, leaf) were used to analyze the expression pattern of the PfMADS-box genes in different tissues. They were taken from a ten-year-old *P. fortunei* tree in June 2021, then frozen in liquid nitrogen for 5 min and stored at −80 °C for RNA isolation.

### 2.2. Identification and Chromosomal Location of MADS-Box Genes in P. fortunei

The genome of *P. fortunei* was acquired from the NCBI database. The identification of MADS-box proteins in *P. fortunei* was carried out by two means. First, the hidden Markov model (HMM) profiles of the SRF-TF (PF00319) and MEF2_binding (PF09047) domains were retrieved from the Pfam database (http://pfam.xfam.org/, accessed on 3 April 2022). Then, MADS-box genes were identified in the *P. fortunei* genome using HMMER (version 3.0) software and the downloaded HMM profiles as the query sequence with the threshold set at e < 1 × 10^−5^. Second, the protein sequences of MADS-box genes in Arabidopsis from the TAIR database (https://www.arabidopsis.org/index.jsp, accessed on accessed on 11 April 2022) were used as query sequences to search against the *P. fortunei* protein dataset using BLASTP with the threshold set at e < 1 × 10^−5^ and 50% identity. The online tools ProtParam (https://web.expasy.org/protparam/, accessed on 20 April 2022) and WoLFPSORT (https://wolfpsort.hgc.jp/, accessed on 20 April 2022) were used to predict the molecular weight, pI, and subcellular location prediction of each protein, respectively.

### 2.3. Evolutionary Relationships among P. fortunei, Oryza sativa, and Arabidopsis thaliana MADS-Box Genes

The protein sequences of the MADS-box of *A. thaliana* and *O. sativa* were retrieved from the TAIR database and Ensembl Plants website (http://plants.ensembl.org/index.html, accessed on 3 May 2022, respectively [7]. MEGA7 was used to construct a phylogenetic tree using neighbor-joining, and bootstrap values were calculated with 1000 repetitions [27].

### 2.4. Synteny Analysis of PfMADS-Box Genes

PfMADS-box genes distributed on all chromosomes were used to analyze the gene replication events within the *P. fortunei* genome. The interspecies analysis of MADS-box genes between *P. fortunei* and *A. thaliana* was performed, selecting *Vitis vinifera* as a bridge. MCscanX (https://github.com/wyp1125/MCScanX.git, accessed on 5 July 2022) was then used to obtain collinearity relationships for each gene pair. The results were visualized by the TBtools (version 1.09868) [28]. The gene cluster definition referred to the report by Holub [29].

### 2.5. Conserved Motif and Gene Structure Analysis

The gene structure information was extracted from the *P. fortunei* genome data, and a gene structure map was drawn using TBtools (version 1.09868) [28]. To identify the conserved motifs in the PfMADS-box proteins, the MEME online tool (https://meme-suite.org/meme/tools/meme, accessed on 7 May 2022) was used with the maximum discovery number of motifs set to 10 [30].

### 2.6. Cis-Regulatory Elements in the Promoters of PfMADS-Box Genes

Generally, the 2000 bp sequence upstream of a gene’s transcriptional start point is considered to be the promoter region [31]. To find responsive elements in the promoters of the PfMADS-box genes, first, the GFF/GTF sequence extractor in TBtools (version 1.09868) was used to obtain their upstream 2000 bp promoter regions of the PfMADS-box genes [28]. Then the online tool PlantCARE (http://bioinformatics.psb.ugent.be/webtools/plantcare/html/, accessed on 12 May 2022) was used to search cis-regulatory elements in these promoter regions.

### 2.7. Expression Profiling of PfMADS-Box Genes

We downloaded the RNA-seq data (PF, PFI, PFIM60-10, PFIM60-2, PFIM20-10, and PFIM20-30) from NCBI (accession number: PRJNA794027) and analyzed the expression of PfMADS-box genes. The heat map was visualized using TBtools (version 1.09868) [28]. Twelve PfMADS-box genes were randomly selected for validation by qRT-PCR.

To understand the roles of PfMADS-box genes in the tissue development of *P. fortunei*, we detected the expression pattern of the genes in three tissues with qRT-PCR experiments.

Total RNA was extracted with the RNA extraction kit (Apexbio, Beijing, China), RNA samples were reverse transcribed using an iScript^TM^ cDNA Synthesis Kit (Bio-Rad, Hercules, CA, USA), and PCR experiments were carried out in triplicate with the CFX96^TM^ Real-Time System (Bio-Rad, Hercules, CA, USA) using iQTM SYBR^®^ Green Supermix (Bio-Rad, Hercules, CA, USA). Gene-specific primers for qRT-PCR are listed in Appendix A. Each sample had three biological replicates. The *PfACTIN1* gene (Pfo06g005670) was used as a reference. The experimental data were analyzed by the 2^−△△Ct^ method [32].

### 2.8. Protein–Protein Interaction Prediction of PfMADS-Box Proteins

The PfMADS-box amino acid sequences were used as targets, and the protein-protein interactions were predicted by the online software STRING (version 11.5, https://string-db.org, accessed on 12 June 2022). The orthologs of *O. sativa* were selected as references. After the BLASTP step, the corresponding proteins with the highest score were used to construct the network.

## 3. Results

### 3.1. Identification and Chromosomal Location of MADS-Box Genes in P. fortunei

A total of 89 MADS-box proteins of *P. fortunei* were identified using the HMM profiles of the SRF-TF (PF00319) and MEF2_binding (PF09047) domains. All PfMADS-box proteins possessed one SRF-TF domain except for *PfMADS34* and *PfMADS36*, which contained two SRF-TF domains in tandem. Fifty-six of these proteins contained a K-box domain. They were named *PfMADS1*–*PfMADS89* in order, according to their localization on the chromosomes (Appendix A). The length of the 89 PfMADS-box proteins ranged from 77 to 460 amino acids, and the molecular weights varied from 8.85 to 53.14 kD. The pI ranged from 4.77 to 10.08: 25 of the proteins were acidic (pI < 6.5), 58 were alkaline (pI > 7.5), and 6 were neutral (pI 6.5–7.5). The subcellular localization prediction of all PfMADS-box proteins indicated that all of them were located in the cell nucleus, which is similar to that of the foxtail millet (*Setaria italica*) [31].

Of the 89 PfMADS-box genes, 84 genes were distributed on 19 chromosomes, and no genes were mapped to Chr12. The gene number on each chromosome ranged from 1 to 16. Chr02 had the most PfMADS-box genes (16), while chr20 contained the least (1). Some MADS-box genes concentrated together in some chromosomal regions, such as 11 genes clustered at the end of chr02. Furthermore, the type Ⅰ and type-Ⅱ PfMADS-box genes are unevenly distributed on the *P. fortunei* chromosomes. The chromosomal positions of five PfMADS-box genes (*PfMADS85*–*PfMADS89*) were not determined because they were on unanchored scaffolds (Figure 1).

### 3.2. Evolutionary Relationships of PfMADS-Box Genes

To investigate the evolutionary relationship of MADS-box family genes among *P. fortunei* (89), *A. thaliana* (107) [7], and *O. sativa* (75) [8], we constructed an unrooted neighbor-joining tree using the MUSCLE sequence alignment (Figure 2). The tree was separated into two core clades based on the classification of *A. thaliana* MADS-box genes, containing 33 type-I and 56 type-II genes, respectively. The 33 type-I PfMADS-box genes were further divided into three subgroups: Mα, Mβ, and Mγ, according to previous reports. Mγ subclade had the most genes (22), and the Mβsubclade had the least number of genes (9). No gene was classified to the Mδ subclade, which differed from that of *Solanum lycopersicum* [33]. Most of the type I MADS-box genes from each of the species clustered into one clade, showing a sister-group relationship. Fifty-six type-II genes were further classified into 50 MIKC^C^-type and 6 MIKC*-type genes. *P. fortunei* appeared to have more MIKC^C^ genes than *A. thaliana* (39) or *O. sativa* (41), which is caused by genetic expansion during evolution. 50 MIKC^C^-type genes were further grouped into 9 *A. thaliana*-specific clades: AGL17 (6), SVP (9), SEP1/2/3 (9), AP1/FUL/CAL (5), FLC/FLM (4), SOC1 (6), AG (6), PI (1), AP3/TT16 (4). The *A. thaliana*-specific subgroup SHP2 had no PfMADS-box members. In particular, the SVP and SEP1/2/3 subclades were significantly expanded in *P. fortunei* (9 SVP and 9 SEP1/2/3 genes) compared with those in *A. thaliana* (4, 6) and *O. sativa* (3, 7). It is noteworthy that most of the genes in the largest subclade Mγ as well as the expanded SEP subclade, were located at the distal telomeric chromosomal regions, which was similar to the results in wheat [9,34].

### 3.3. Synteny Analysis of PfMADS-Box Genes

Synteny comparison analysis is helpful in understanding the evolutionary and functional relationships of gene family members. Seventeen gene duplication events among PfMADS-box genes were discovered, and 16 of them were subclade-specific (Figure 3a). The findings implied that some PfMADS-box genes were created by segmental duplication, which would have resulted in multiple homologs in distinct chromosomes. Almost all duplicated gene pairs were located on distal telomeric chromosomal segments, which was consistent with the finding that recombination events occurred at high frequency in subtelomeric segments [35]. Moreover, interspecific collinearity analysis between *P. fortunei* and the model plant A. thaliana was performed selecting the V. vinifera genome as a bridge. When compared to V. vinifera, the genomes of *P. fortunei* and *A. thaliana* have undergone one cycle and two cycles of whole-genome duplication, respectively (Appendix A) [25,36]. We found that 20 of the MADS-box genes in *P. fortunei* were related to 27 genes in *A. thaliana* (Figure 3b). The results showed that the divergence was generated during the diploidization after the polyploidization.

### 3.4. Gene Structure and Conserved Motif Composition of PfMADS-Box Genes

The intron-exon patterns were analyzed to explore the structural diversity of the PfMADS-box genes. As shown in Appendix A, the number of exons in the PfMADS-box genes ranged from 1 to 16. The distribution of exons was different in type I and type II PfMADS-box genes (Appendix A). Most of the type I genes had no introns or only one intron, and the sequence length was relatively short; the exceptions were *PfMADS9* and *PfMADS80*, which had more than one intron. All type II genes had more than 5 introns, and the sequence is generally longer. These intron-exon distribution patterns were similar to those reported in *A. thaliana* and *O. sativa* [7,8].

The conserved motifs of 89 PfMADS-box proteins were determined by using the MEME online tool. Ten conserved motifs were identified and named motif 1 to motif 10 (Figure 4). PfMADS-box proteins from the same subfamily showed similar patterns of conserved motifs. Motif 2, as a part of the SRF-TF domain, was the most frequently present in all 89 PfMADS-box proteins, and the following is motif 1, which also be included in the SRF-TF domain. As anticipated, some motifs were specific to the subfamily, Motif 3 and motif 6, which were annotated as the K-box domain, and occurred specifically in type Ⅱ proteins. Motif 7 and motif 8 were predicted only in the type Ⅰ proteins, and motif 10 occurs only in the SVP subfamily. However, we discovered that some members located on the distal telomeric regions, such as *PfMADS29*, *PfMADS34*, *PfMADS50*, *PfMADS63*, *PfMADS64*, and *PfMADS82*, have or do not have the motif 4, which differed from other members of the same subfamilies. These findings indicated several PfMADS-box genes underwent rapid evolution, and a similar situation exists in foxtail millet [31].

### 3.5. Cis-Regulatory Elements in the Promoters of PfMADS-Box Genes

The presence of cis-regulatory elements in the promoter region of the gene is crucial for gene expression regulation. To explore the potential CREs present in the PfMADS-box gene family, the PlantCARE website was used to search against the 2000 bp sequence upstream from the coding region of each PfMADS-box gene. A total of 25 types of cis-acting elements (2026) were found (Appendix A). According to their function, most cis-elements were classified into three groups: growth and biological process-sensitive elements (1370), stress-responsive elements, and hormone-responsive elements. The light-responsive element (77.66%) was the most common, present in all PfMADS-box genes, indicating that PfMADS-box expression was induced by light. The MeJA-responsive element (35.06%) was the most prevalent hormone-responsive element, followed by the abscisic-responsive element (33.61%), showing that the expression of PfMADS-box genes was associated with phytohormone. The number of elements engaged in defense and stress response was the greatest among the five stress response elements (34.29%), followed by low-temperature responsive elements (30.00%) and drought-responsive elements (27.14%), demonstrating the existing response of PfMADS-box genes to abiotic stress in *P. fortunei* (Figure 5). We also noticed that some elements were particular to several PfMADS-box genes, such as cell cycle regulatory elements found only in *PfMADS5*, *PfMADS58*, *PfMADS60*, and *PfMADS75*, and photoresponsive elements found only in *PfMADS10*, as well as root-specific elements found only in *PfMADS10* and *PfMADS66*, implying the specificity of certain PfMADS-box genes involved in plant growth and development.

### 3.6. Tissue Expression Profiles of PfMADS-Box Genes

Sixteen genes homologous to the Arabidopsis ABCDE model were selected to detect the expression of PfMADS-box genes in three different tissues (flower, stem, and leaf) (Figure 6a). Transcripts of seven genes were the most abundant in flowers, suggesting that these genes might play key roles in the regulation of floral organogenesis. This phenomenon has also been observed in rice [8] and wheat [9]. *PfMADS44* was only expressed in the stem, implying its function in regulating internode development. Additionally, the expression levels of *PfMADS3*, *PfMADS22*, *PfMADS45*, *PfMADS10*, *PfMADS26*, and *PfMADS27* were higher in the leaf than in the other two tissues. The PfMADS-box gene expression profiles in three tissues indicated their functional divergence or sub-functionalization.

### 3.7. Expression of PfMADS-Box Genes in Response to PaWB Phytoplasma Infection

To investigate the response of PfMADS-box genes to phytoplasma infection, we analyzed the expression pattern of these genes in PF, PFI, and MMS-treated PFI seedlings. After the treatment with MMS (60 mg·L^−1^ for 20 days, 20 mg·L^−1^ for 30 days), the infected seedlings exhibited recovery, while PaWB phytoplasma could be detectable in the samples treated by low-concentration MMS, but not in those treated by high-concentration MMS [37]. In PFI vs. PF, 46 PfMADS-box genes exhibited differential expression, 42 genes were down-regulated, and 4 genes were up-regulated (Figure 6b). In the high-concentration group (PF, PFI, PFIM60-10, PFIM60-20), 10 genes (*PfMADS3*, *PfMADS13*, *PfMADS22*, *PfMADS49*, *PfMADS54*, *PfMADS57*, *PfMADS59*, *PfMADS64*, *PfMADS77*, and *PfMADS87*) were screened out from the 46 genes, and their expression was up-regulated (or down-regulated) in PFI vs. PF, as well as down-regulated (or up-regulated) in both of PFIM60-10 vs. PFI and PFIM60-20 vs. PFIM60-10. Likewise, seven genes of 46 (*PfMADS3*, *PfMADS13*, *PfMADS48*, *PfMADS54*, *PfMADS57*, *PfMADS79*, and *PfMADS87*) were obtained in the low-concentration group (PF, PFI, PFIM20-10, PFIM20-30). The intersection included five MADS-box genes: *PfMADS3*, *PfMADS13*, *PfMADS54*, *PfMADS57*, and *PfMADS87*. These five MADS-box genes, the expression levels of which were restored gradually with the extension of treatment duration, were considered to play roles in the incidence of arbuscular illness. Interestingly, these five genes belonged to the type Ⅱ of the MADS-box gene, implying that the type-II MADS-box genes may be particularly significant for PaWB occurrence. QRT-PCR validation showed that the expression trends of 12 genes selected randomly were consistent with those from RNA-seq data (Figure 6c), which confirmed the reliability of the RNA-seq data.

*OsMADS18* and *OsMADS57* were found to have functions in plant tillering [38,39], which were highly homologous with *PfMADS3* and *PfMADS54*, respectively. So, we used the STRING website to construct the protein interaction networks of *PfMADS3* and *PfMADS54*, selecting rice as a reference. The results showed that they might interact with other PfMADS-box proteins, which is in line with previous reports that MADS-box proteins activate or repress target genes by forming homodimers or heterodimers [40]. And beyond that, *PfMADS3* could act with FL protein (Pfo14g003770.1), while *PfMADS54* could work with D14 (Pfo11g010150.1) and TB1(Pfo17g002050.1) (Figure 7).

## 4. Discussion

PaWB is a serious disease of Paulownia, which greatly reduces the growth and biomass of *P. fortunei*. In recent years, some genes related to the occurrence of PaWB have been discovered [25,41,42,43], including the MADS-box gene, but there is no systematic report on the MADS-box family of *P. fortunei.* Therefore, the focus of this study was to explore the PfMADS-box gene’s probable molecular functions in the development of PaWB. A total of 89 PfMADS-box gene family members located on 19 chromosomes were found in the genome of *P. fortunei*. Collinear analysis showed that fragment duplication events of MADS-box genes occurred not only among *P. fortunei* members but also between *P. fortunei* and *A. thaliana*. The tissue expression patterns of some genes indicated that the PfMADS-box genes might undergo subfunctional differentiation. Furthermore, we employed RNA-seq data to investigate the response of PfMADS-box genes to phytoplasma infection and found several PfMADS-box genes that may be related to the occurrence of PaWB. The identification and function prediction of the PfMADS-box genes will provide a basis for the future understanding of their functions associated with the PaWB pathogenesis mechanism.

In this study, we identified 89 MADS-box gene family members in the *P. fortunei* genome. In comparison with previous studies, we found that the number of MADS-box genes varies in different species: *A. thaliana* (107) [7], *Populus trichocarpa* (101) [44], *Medicago sativa* (120) [45], *Malus domestica* (146) [46], and *S. lycopersicum* (131) [33]. This suggested that *P. fortunei* might have lost some MADS-box genes during evolution. The number of Mα PfMADS-box genes (2) is much less than *A. thaliana* (11) and *O. sativa* (10), and none of the PfMADS-box gene members belongs to the SHP2 subclade. Therefore, we inferred that Mα- and SHP2- MADS-box genes were lost during the evolution of *P. fortunei*. Moreover, SEP1/2/3, SVP, and AG subclades have increased significantly, which may ascribe gene segmental duplication.

The structural diversity of genes drives the evolution of multigene families. Intron loss and insertion mutations have been shown to be common during the evolution of plant MADS-box genes [47,48]. In this study, the number of introns in the PfMADS-box genes varied greatly; most of the type Ⅰ PfMADS-box genes had no introns or had a single intron, whereas the MIKC^C^ genes had from 5 to 12 introns. The patterns of exon-intron structures in type I and type II genes are conserved across diverse plant taxa, including Arabidopsis [7], rice [8], and grape [49], indicating that MADS-box transcription factors are highly conserved among plants. Additionally, the PfMADS-box protein motifs of the same subfamily were not identical, which suggested that the loss or gain of introns may be a pattern of PfMADS-box gene evolution and could be a major contributing factor to the functional diversity of the PfMADS-box family.

Proliferating branches and plant dwarfing are typical symptoms of phytoplasma disease [50]. It has been shown that MADS-box genes could be involved in plant branching control by participating in the strigolactone (SL) synthetic or signaling pathways [51,52]. In transgenic rice lines overexpressing *OsMADS57*, tillering increased [51]; in *OsMADS18* mutant rice plants, axillary buds increased [9]. *OsMADS57* can repress the SL receptor gene *D14* by binding to its promoter and interacting with *OsMADS18* or OsTB1 [39,53]. Therefore, the *OsMADS18*–*OsMADS57* interaction is involved in ABA and SL-regulated growth in rice. *PfMADS3* and *PfMADS54* were found to be highly homologous with *OsMADS18* and *OsMADS57,* respectively. The expression of *PfMADS54* and *PfMADS3* was upregulated when the *P. fortunei* seedlings were infected by PaWB phytoplasma and decreased with increased MMS treatment time. We speculated that *PfMADS54* might interact with *PfMADS3* or PfTB1 (Pfo17g002050.1) to inhibit the transcription of the downstream gene *PfD14* (Pfo11g010150.1), which resulted in the uncontrolled growth of auxiliary buds in infected plants. In addition, *PkMADS1* has been reported as a regulator of shoot morphogenesis [20], high homology (97% identity) with *PfMADS57* (Appendix A), and the expression level of *PfMADS57* was increased in PFI compared with PF, and then returned in PFI after MMS treatment. Thus, we presumed *PfMADS57* functioned during the pathogenesis of PaWB.

Ubiquitination has been demonstrated to play a vital role in plant disease occurrence. In tobacco, the effector βC1 could interact with the RING E3 ligase NtRFP1 and direct its degradation depending on the ubiquitin/26S proteasome system, which reduces the severity of βC1-induced symptoms [54]. The phytoplasma effector SAP54 targets the AtMADS-box proteins (SEP3, SOC1, and AP1) and degrades them in a RAD23-dependent manner, which causes abnormal floral organ development [23]. Therefore, we hypothesized that *PfMADS3*, as a homolog of SOC1, not only functioned in flower development but also regulated the formation of axillary buds. PfMADS3 might be degraded in a ubiquitylation-dependent by PaWB phytoplasmas. Together, these results suggest that *PfMADS3*, *PfMADS54*, and *PfMADS57* respond to the infection of witches’ broom phytoplasma and may be related to axillary clusters caused by PaWB.

In summary, we identified 89 MADS-box gene family members in the *P. fortunei* genome and studied their evolutionary connection, conserved motifs, gene structure, and cis-acting elements, as well as analyzed the relationship between PfMADS-box gene expression and occurrence of PaWB. Our findings lay the foundation for a comprehensive functional characterization of the MADS-box gene family in *P. fortunei* and provided candidate genes for studying the role of the MADS-box gene family in the formation of PaWB. Additionally, the potential functions and regulatory mechanisms of *PfMADS3*, *PfMADS54*, and *PfMADS57* will be experimentally illuminated in our further research.

## Figures and Tables

**Figure 1 genes-14-00696-f001:**
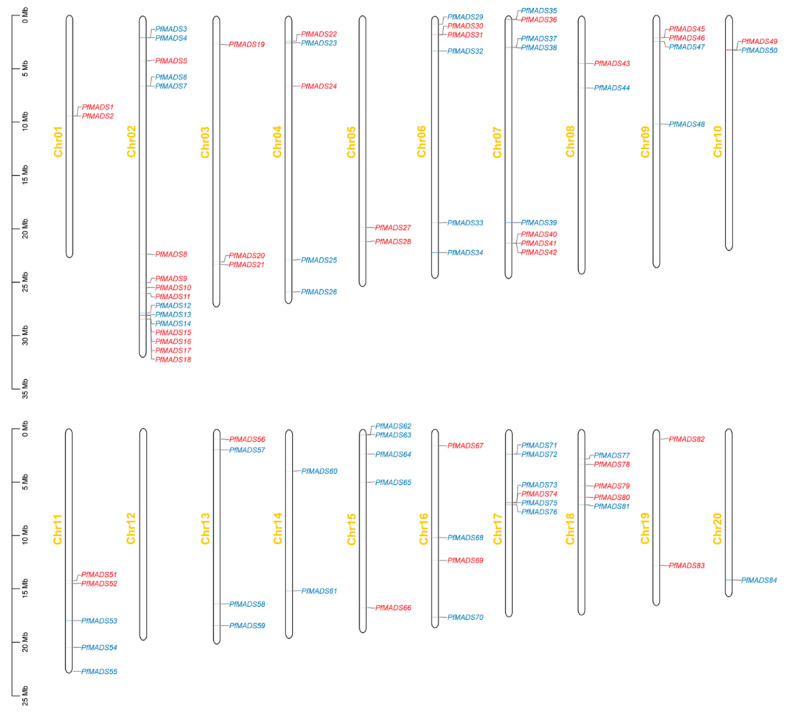
Chromosomal location of PfMADS-box genes in *P. fortunei*. The scale of the chromosome is millions of base pairs (Mb). Type Ⅰ and type Ⅱ genes are colored red and blue, respectively.

**Figure 2 genes-14-00696-f002:**
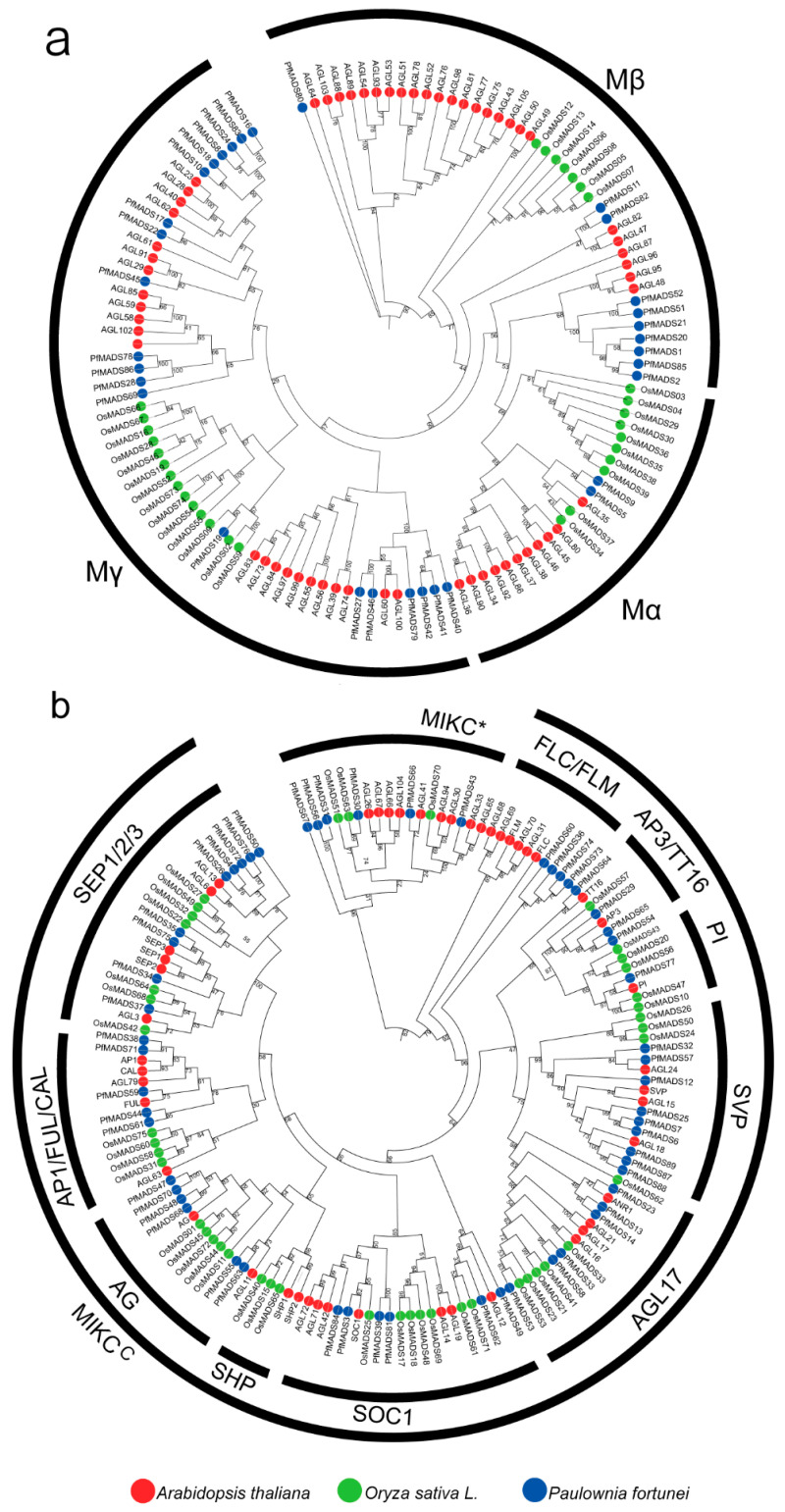
Phylogenetic analysis of type I (**a**) and type II (**b**) MADS-box proteins from *P. fortunei*, *A. thaliana,* and *O. sativa.* The MADS-box genes are labeled in red, green, and blue for *A. thaliana, O. sativa*, and *P. fortunei,* respectively.

**Figure 3 genes-14-00696-f003:**
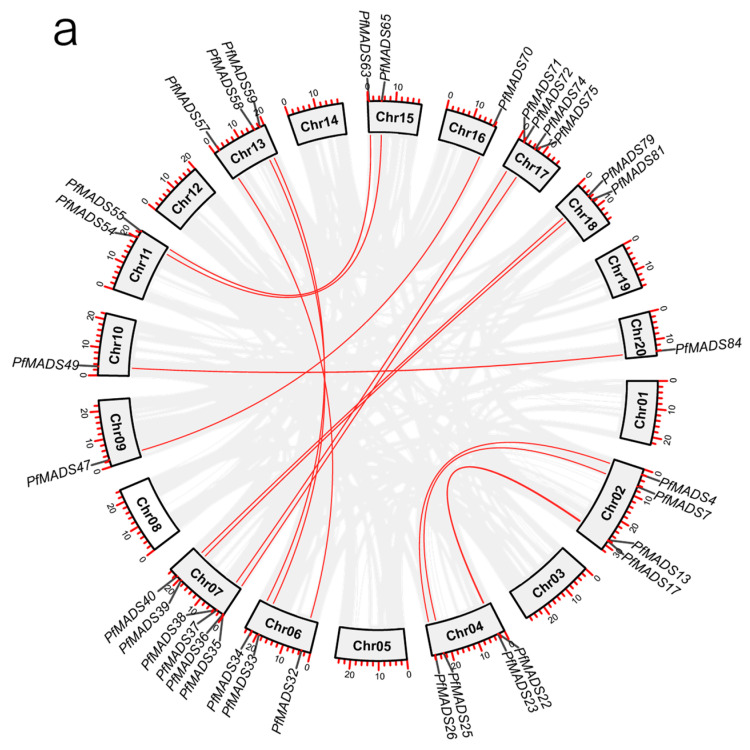
Synteny analysis of MADS-box genes in *P. fortunei*. (**a**) The duplication relationship of PfMADS-box genes. Gray lines represent all synteny blocks in the *P. fortunei* genome, red lines indicate the duplicated PfMADS-box gene pairs. (**b**) homologous analysis of MADS-box genes between *P. fortunei* and *A. thaliana*.

**Figure 4 genes-14-00696-f004:**
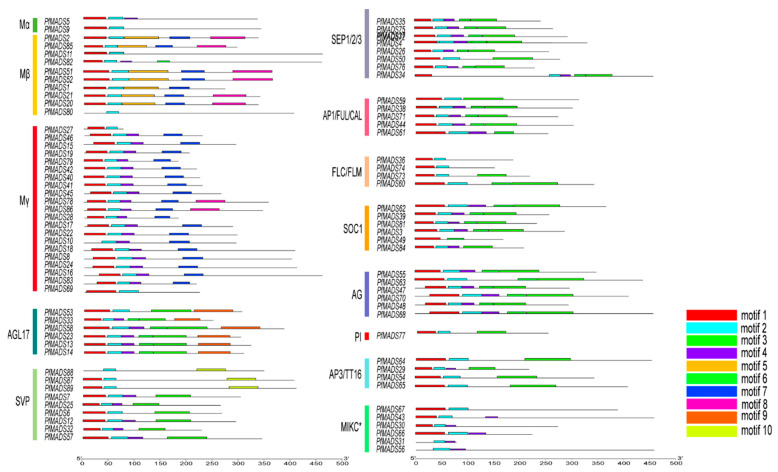
The motif composition of PfMADS-box genes. Ten motifs are labeled by different colors.

**Figure 5 genes-14-00696-f005:**
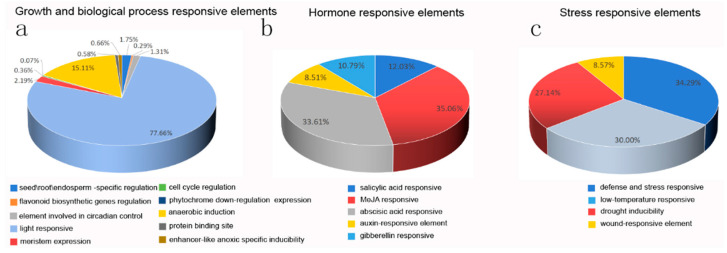
Percentage distribution of cis-regulator elements in the promoters of PfMADS-box genes based on the putative functions. (**a**) Growth and biological process-responsive elements. (**b**) Stress-responsive elements. (**c**) Hormone-responsive elements.

**Figure 6 genes-14-00696-f006:**
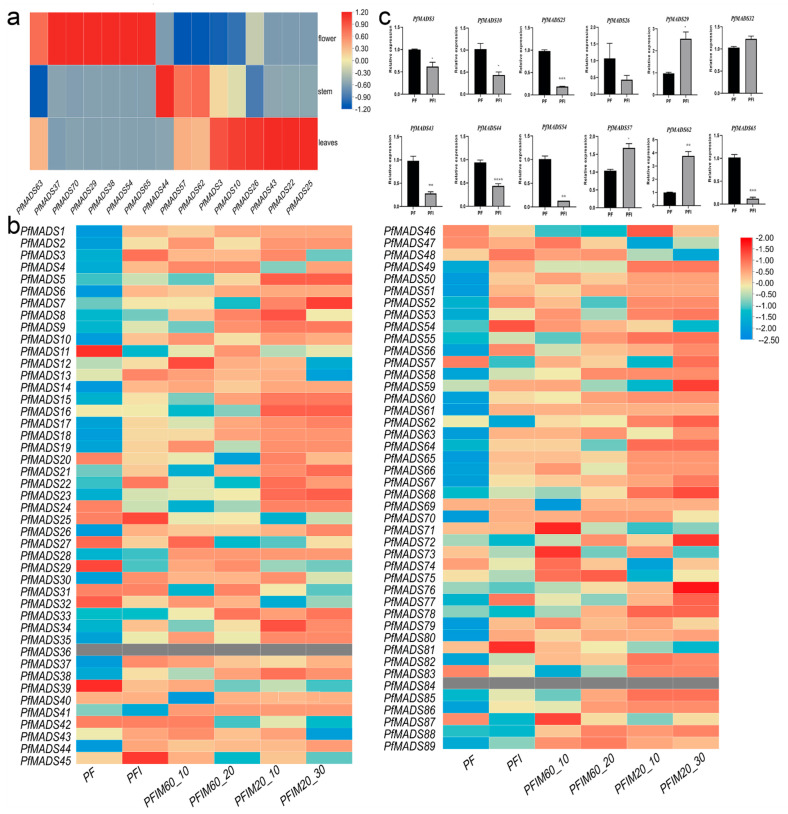
Expression pattern analysis of PfMADS-box genes. (**a**) heat map of PfMADS-box genes in PF, PFI, and MMS-treated PFI seedlings. PFIM60-10: PFI treated with 60 mg·L^−1^ MMS for 10 days; PFIM60-20: PFI treated with 60 mg·L^−1^ MMS for 20 days; PFIM20-10: PFI treated with 20 mg·L^−1^ MMS for 10 days; PFIM20-30: PFI treated with 20 mg·L^−1^ MMS for 30 days. (**b**) Expression levels of several PfMADS-box genes in different tissues. (**c**) Results of qRT-PCR validation. Values are means ± SD (*n* = 3). Asterisks indicate a significant difference (****, *p* < 0.0001; ***, *p* < 0.001; **, *p* < 0.01; *, *p* < 0.05).

**Figure 7 genes-14-00696-f007:**
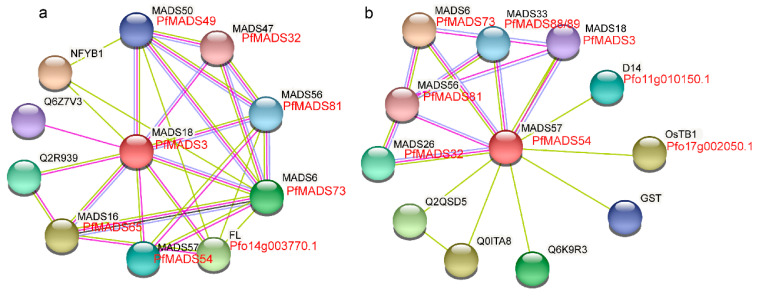
Protein–protein interaction network of PfMADS3 (**a**) and PfMADS87 (**b**). Circle nodes in the network represent proteins, and straight lines connecting the nodes represent interaction relationships. Indigo blue and purple lines indicate known interactions; green, red, and navy blue balls indicate predicted interactions; yellow, black and pearl blue indicate texting, co-expression, and protein homology interactions, respectively.

## Data Availability

MADS-box gene members of A. thaliana were downloaded from the TAIR database (http://www.arabidopsis.org, accessed on 6 May 2022). MADS-box gene members of O. sativa were downloaded from Ensembl Plants website (http://plants.ensembl.org/index.html, accessed on 3 May 2022). A total of 18 transcriptome datasets of P. fortunei were deposited in the Genome Sequence Archive (SRA) database of NCBI under the accession number PRJNA794027 (accessed on 29 June 2022).

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
