# Peer review of "Genome-Wide Identification and Expression of the *Paulownia fortunei* MADS-Box Gene Family in Response to Phytoplasma Infection"

_genes, 2023, doi:10.3390/genes14030696_

Round 1

Reviewer 1 Report

The authors have done the genome wide analysis of the MADS gene family from Paulownia fortunei and their involvement in response to phytoplasma infection. The authors have analysed the evolution of MADS box genes in relation to Arabidopsis and Rice. The expression of some MADS box genes in response to phytoplasma infection is studied. However the main focus of the present study is on the MADS box gene family in Paulownia fortunei. 

As the tree belongs to Asterids the authors could do some comparison with related tree species for the evolutionary analysis. 

The authors also need to discuss the role of MADS box genes during phytoplasma infection. If genes are getting upregulated then how are they acting? What downstream genes are being regulated?

Author Response

Dear editor and reviewers:
We would like to thank you and the reviewers for your careful reading, comments and constructive suggestions, and also thank you for giving us an opportunity to revise our manuscript entitled “Genome-wide identification and expression of the Paulownia fortunei MADS-box gene family in response to phytoplasma infection [genes-2189791]. Here, we have revised our manuscript according to your suggestions, and our final version has been proofread by a native English professional with science background.Below are our point-by-point responses:

The authors have done the genome wide analysis of the MADS gene family from Paulownia fortunei and their involvement in response to phytoplasma infection. The authors have analysed the evolution of MADS box genes in relation to Arabidopsis and Rice. The expression of some MADS box genes in response to phytoplasma infection is studied. However the main focus of the present study is on the MADS box gene family in Paulownia fortunei. 

As the tree belongs to Asterids the authors could do some comparison with related tree species for the evolutionary analysis. 

Response: Thank you for your constructive suggestion! A total of 275 reference genomes of Asterids, most of which are herbadeous plants, are deposited in NCBI genome database. We selected five tree species from them. We did not successfully construct phylogenetic trees of Paulownia fortunei with the reported MADS gene families of Camellia chekiangoleosa (Zhou et al., 2023) or Camellia sinensis (Zhang et al., 2021). MADS gene families of other Asterids woody plants, such as Fraxinus Americana, Olea europaea and Osmanthus fragrans, have not been identified so far. I have to cancel the evolutionary analysis. I am very sorry for this.

Zhou, P., Qu, Y., Wang, Z., Huang, B., Wen, Q., Xin, Y., Ni, Z., & Xu, L. (2023). Gene Structural Specificity and Expression of MADS-Box Gene Family in Camellia chekiangoleosa. International journal of molecular sciences, 24(4), 3434. https://doi.org/10.3390/ijms24043434

Zhang, Z. B., Jin, Y. J., Wan, H. H., Cheng, L., & Feng, Z. G. (2021). Genome-wide identification and expression analysis of the MADS-box transcription factor family in Camellia sinensis. Journal of applied genetics, 62(2), 249–264. https://doi.org/10.1007/s13353-021-00621-8

The authors also need to discuss the role of MADS box genes during phytoplasma infection. If genes are getting upregulated then how are they acting? What downstream genes are being regulated?

Response: Thank you for your constructive suggestion! We focused on two genes that be highly homologous with OsMADS18 and OsMADS57: PfMADS54 and PfMADS3, their expression was upregulated when the P. fortunei seedlings were infected by PaWB phytoplasma, and decreased with increased MMS treatment time. They might interact to regulate the downstream gene PfD14 (Pfo11g010150.1), which resulted in the occurrence of PaWB. We added some sentences for this in Discussion section (Line 371-384).

Reviewer 2 Report

The authors identified 89 MADS-box genes spread across 14 gene families in Paulownia fortune and attempted to describe their role in PaWB infection. Their study narrowed down the involvement of 3 genes in PaWB. The methods are very well explained. Similar studies have been done in other plant species but not in Paulownia fortune. Although the study is well-defined but I have some significant concerns outlined below:

1. Although, I understand that it takes at least 4-5 years for this plant to flower but taking samples from 10 years old plant which was not grown in a controlled environment, may skew the results. The spatial expression of the tested genes may be influenced by several factors due to an uncontrolled environment. 

2. The description of the role of MMS in the treatment of phytoplasma infection in the methods section is inappropriate. It could be described briefly in the results section to score the rationale for using MMS-treated samples in their expression studies.

Although authors have cited A. thaliana and O. sativa all across the manuscript, but they failed to cite recent papers on potato and wheat on similar lines-

1.     Wang Y, Zhang J, Hu Z, Guo X, Tian S, Chen G. Genome-Wide Analysis of the MADS-Box Transcription Factor Family in Solanum lycopersicum. Int J Mol Sci. 2019 Jun 18;20(12):2961. doi: 10.3390/ijms20122961. PMID: 31216621; PMCID: PMC6627509.

2.     Raza Q, Riaz A, Atif RM, Hussain B, Rana IA, Ali Z, Budak H, Alaraidh IA. Genome-Wide Diversity of MADS-Box Genes in Bread Wheat is Associated with its Rapid Global Adaptability. Front Genet. 2022 Jan 17;12:818880. doi: 10.3389/fgene.2021.818880. PMID: 35111207; PMCID: PMC8801776.

How is this study any different from the above two?

3. The description of the role of MMS in the treatment of phytoplasma infection in the methods section is inappropriate. It could be described briefly in the results section to score the rationale for using MMS-treated samples in their expression studies.

4. These proteins display a range of properties: small molecular weight to high molecular weight, acid to basic nature. It would be good if authors could highlight this variability in terms of functions. For example, do proteins with low molecular weight and acidic in nature have a specific function different from those with high molecular weight and basic in nature? 

5. Type II PfMADS genes are located mostly in the telomeric region. Can authors comment on the role of these genes in telomere expansion or is there any study in any plant species highlighting the chromosomal location of these genes?

6. The evolutionary relationship is explained well. However, I suggest a minor improvement in the figure to show MIKCctype like they’ve shown MIKC* subfamily of type II PfMADS.

7. Why V. vinifera genome was used as a bridge in synteny analysis is not clear.

8. In lines 26-263 “However, we discovered that the motif composition of some members, including PfMADS29,PfMADS30, PfMADS34, PfMADS36, and PfMADS82, differed from that of other members of the same subfamily, which could be due to the PfMADS-box gene evolution, and a similar situation exists in foxtail millet”, authors must explain the difference in motif composition here.

9. Effect of deletion of some of the PfMADS genes on plant growth needs to be validated. For example, the role of PfMADS44 in internode development could be studied by checking the phenotype of plant lacking PfMADS44 gene. Phenotypic studies could bolster the claims made by authors based solely on bioinformatics analysis.

10. I am not very convinced by their protein-protein interaction networks as there are plenty of evidences across literature that shows that homologs may differ in functions as well as interactomes. So, using rice as a reference may not be sufficient. The authors could use other references as well to show that the interaction is conserved across organisms and then extrapolate this to Paulownia fortune.

11. What about the other genes in transcriptome analysis? I understand that other genes are not necessary in this study but it should be mentioned at least that other genes are also differentially expressed in transcriptomics study.

12. The conclusions drawn in the study are too optimistic in absence of any experimental evidence.

13. Also a minor suggestion: Fig 6b should be 6a and vice-versa based on the appearance of the result in the text. There are several grammatical errors in the manuscript.

Reviewer 3 Report

Updates to the references are required. 

Author Response

Dear editor and reviewers:
We would like to thank you and the reviewers for your careful reading, comments and constructive suggestions, and also thank you for giving us an opportunity to revise our manuscript entitled “Genome-wide identification and expression of the Paulownia fortunei MADS-box gene family in response to phytoplasma infection [genes-2189791]. Here, we have revised our manuscript according to your suggestions, and our final version has been proofread by a native English professional with science background.Below are our point-by-point responses:

Comments and Suggestions for Authors

Updates to the references are required. 

Response: Thank you for your comment. We cited some recently published articles and corrected the format of some references according to your suggestion.

Round 2

Reviewer 2 Report

The manuscript has substantially improved. I am satisfied with the author's response. Fig 6 shows 'b' and 'a' in reverse order. With changing the order, I meant change make 6b as 6a and vice-versa. Please correct it.

Author Response

Dear  reviewer:
We want to thank you for your constructive suggestions, and also for allowing us  to revise our manuscript entitled “Genome-wide identification and expression of the Paulownia fortunei MADS-box gene family in response to phytoplasma infection ” [genes-2189791]. Here, we have adjusted the order of the three pictures in Figure 6 according to your suggestions.
